# Development, feasibility and potential effectiveness of community-based continuous mass dog vaccination delivery strategies: Lessons for optimization and replication

**Christian Tetteh Duamor**[1,2,3]*, **Katie Hampson**[3], **Felix Lankester**[4,5], **Ahmed Lugelo**[6], **Emmanuel Mpolya**[1], **Katharina Kreppel**[1], **Sarah Cleaveland**[3], **Sally Wyke**[7]

**1** Department of Global Health and Biomedical Sciences, School of Life Sciences and Bioengineering, Nelson Mandela African Institution of Science and Technology, Arusha–Tanzania, **2** Environmental Health and Ecological Sciences Thematic Group, Ifakara Health Institute–Tanzania, **3** Boyd Orr Centre for Population and Ecosystem Health, Institute of Biodiversity, Animal Health & Comparative Medicine, College of Medical, Veterinary and Life Sciences, University of Glasgow, Glasgow, United Kingdom, **4** Paul G. Allen School for Global Health, Washington State University, Pullman, Washington, United States of America, **5** Global Animal Health Tanzania, Arusha, Tanzania, **6** Sokoine University of Agriculture, Morongoro–Tanzania, **7** School of Social and Political Sciences, School of Health and Wellbeing, College of Social Sciences, University of Glasgow, Glasgow, United Kingdom

* tettehcd@outlook.com

## Abstract

### Objectives

Dog vaccination can eliminate rabies in dogs, but annual delivery strategies do not sustain vaccination coverage between campaigns. We describe the development of a community-based continuous mass dog vaccination (CBC-MDV) approach designed to improve and maintain vaccination coverage in Tanzania and examine the feasibility of delivering this approach as well as lessons for its optimization.

### Methods

We developed three delivery strategies of CBC-MDV and tested them against the current annual vaccination strategy following the UK Medical Research Council's guidance: i) developing an evidence-based theoretical framework of intervention pathways and ii) piloting to test feasibility and inform optimization. For our process evaluation of CBC-MDV we collected data using non-participant observations, meeting reports and implementation audits and in-depth interviews, as well as household surveys of vaccination coverage to assess potential effectiveness. We analyzed qualitative data thematically and quantitative data descriptively.

### Results

The final design included delivery by veterinary teams supported by village-level one health champions. In terms of feasibility, we found that less than half of CBC-MDV's components

**Data Availability Statement:** All relevant data are within the manuscript and its Supporting Information files.

**Funding:** Funding for postgraduate study (CTD) and supervision (KK) was received from the DELTAS Africa Initiative [Afrique One-ASPIRE /DEL-15-008]. Afrique One-ASPIRE (URL: www. http://afriqueoneaspire.org/) is funded by a consortium of donors, including the African Academy of Sciences (AAS) (URL: www.http://www.aasciences.africa/), Alliance for Accelerating Excellence in Science in Africa (AESA), the New Partnership for Africa's Development Planning and Coordinating (NEPAD) Agency, the Wellcome Trust [107753/A/15/Z] and the UK government. The mass dog vaccination and research activities were funded by the Department of Health and Human Services of the National Institutes of Health [R01AI141712] (FL, SC, KH & SW). (URL: www. https://www.nih.gov/about-nih) Research activities were funded by MSD Animal Health who also donated the dog vaccines. The Wellcome Trust funded KH and CTD [207569/Z/17/Z]. URL: www. https://wellcome.org/ The funders had no role in study design, data collection and analysis, decision to publish, or preparation of the manuscript.

**Competing interests:** The authors have declared that no competing interests exist.

were implemented as planned. Fidelity of delivery was influenced by the strategy design, implementer availability and appreciation of value intervention components, and local environmental and socioeconomic events (e.g. elections, funerals, school cycles). CBC-MDV activities decreased sharply after initial campaigns, partly due to lack of supervision. Community engagement and involvement was not strong. Nonetheless, the CBC-MDV approaches achieved vaccination coverage above the critical threshold (40%) all-year-round. CBC-MDV components such as identifying vaccinated dogs, which village members work as one health champions and how provision of continuous vaccination is implemented need further optimization prior to scale up.

## Interpretation

CBC-MDV is feasible to deliver and can achieve good vaccination coverage. Community involvement in the development of CBC-MDV, to better tailor components to contextual situations, and improved supervision of activities are likely to improve vaccination coverage in future.

## Author summary

Annual mass dog vaccination campaigns that reach at least 70% of the dog population, should maintain sufficient herd immunity (sustain vaccination coverage above 40%) between campaigns to interrupt rabies transmission. However, it is often challenging to reach 70% of the dog population with annual vaccination campaigns. We hypothesized that a community-based continuous approach to dog vaccination could better maintain high levels of vaccination coverage all-year-round. We describe the development of a community-based continuous approach to dog vaccination in Tanzania, and assessed the feasibility of delivering its components, its potential effectiveness and lessons for its optimization. We found that the approach was well accepted, as its development involved key stakeholders. Although less than half of the components of the community-based continuous approach were delivered exactly as planned, over 70% of dogs were vaccinated and the approach maintained coverage above the critical vaccination threshold throughout the year. We conclude that it is feasible to deliver a community-based continuous approach to dog vaccination, but that some components need further improvement; more supervision and community involvement should lead to better outcomes.

## Introduction

Rabies is a central nervous system infection that can infect all mammals. The disease has a case fatality rate approaching 100% [1,2]. Globally, human deaths are estimated at about 59,000 per annum, with 99% due to domestic dog-mediated transmission [3]. The burden of rabies is highest in endemic regions where both human and animal rabies vaccines are not reliably accessible [3,4].

Rabies is controllable for several reasons: domestic dogs are the primary source of infections to humans and rabies has a consistently low basic reproductive number ($R_0 < 2$) across a wide range of settings [1]; dogs in endemic regions are typically accessible for vaccination [5,6]; and the low $R_0$ means that the critical vaccination threshold required to achieve herd immunity is

relatively low (approximately 40%). Despite these reasons, rabies remains endemic in many settings and only limited dog vaccination is undertaken. A possible concern should mass dog vaccination be scaled up is that, despite the critical vaccination threshold being low, to sustain vaccination coverage *above* this level over the course of the year, annual vaccination campaigns must reach a higher proportion of dogs, of around 70% [1,7,8].

Most endemic countries including Tanzania, where mass dog vaccination (MDV) has been initiated, use annual team-delivered approach in which government vaccination teams use cold-chain stored vaccines to conduct annual vaccination clinics in targeted villages. However, annual team-delivered approach (subsequently referred to in this study as the pulse approach) is affected by several factors that limit its ability to achieve and sustain vaccination coverages above the critical threshold to control rabies. These include: high rates of dog population turn-over in most endemic countries, which results in rapid declines in population immunity in the interval between annual campaigns [9,10]; teams needing to travel long distances on dirt roads from cold chain facilities, which is sometimes possible only at certain times of the year; campaign day(s) being negatively affected by agricultural cycles, inclement weather, school days, funerals, and local festivals [8]; high fixed vehicle and personnel costs, with the cost-per-dog vaccinated reaching as high as $7.36 [11–13].

Recent research has shown that Nobivac Canine Rabies Vaccine, a widely used vaccine for dogs [14], is thermotolerant and can induce equivalent immune responses following non cold chain storage at temperatures up to 30˚C for three months. Thermotolerance remained when the vaccine was stored in rural Tanzania in locally made passive cooling devices, within which temperatures were kept between 18–20˚C despite ambient temperatures reaching 37˚C [15]. This research has created opportunities for new approaches to rabies vaccine distribution and delivery, including options for the storage of vaccines in remote communities which would allow all year-round routine vaccination of dogs by community-based personnel. A community-based continuous mass dog vaccination (CBC-MDV) approach has the potential to sustain population immunity above the critical threshold, as new puppies and other susceptible dogs (for example, newly acquired dogs or those that missed previous vaccination campaigns) can be vaccinated without having to wait for the annual campaign. Through empowering communities to own and sustain local dog vaccination efforts, it has been hypothesized that a CBC-MDV model could also result in more dogs being reached at less cost per animal vaccinated [14,16,17].

CBC-MDV is a complex intervention, with several interacting components such as the involvement of local veterinary authorities and communities, local storage of dog rabies vaccines outside of the cold chain system and a continuous approach to dog vaccine delivery. Consequently, the intervention could operate differently in different settings. The UK Medical Research Council Guidance on developing and evaluating complex interventions prior to full scale evaluation recommends a systematic approach to intervention development [18]. This approach should include the development of the intervention with stakeholders, a theoretical understanding of how it is likely to operate, and piloting of its delivery with a view to evaluating the feasibility of its delivery in the long term, prior to full scale evaluation. Following the UK Medical Research Council Guidance, we describe the formative work in developing three delivery strategies of CBC-MDV and evaluation of the feasibility of delivering its strategy components. We also assess the potential of CBC-MDV models to sustain vaccination levels above the critical threshold for rabies elimination and lessons for its optimization and replication.

## Methods

### Ethics statement

The study was approved by the Institutional Animal Care and Use Committee, Washington State University [Approval No. 04577–001], the Tanzania National Medical Research Institute [NIMR/HQ/R.8a/Vol.IX/2788] and Ifakara Health Institute [IHI/IRB/No:024–2018].

### Study design

The research was conducted in two stages, Fig 1 provides a schematic overview of the processes involved.

### Phase 1: Developing components of CBC-MDV

CBC-MDV was developed to be delivered in rural Tanzania and piloted in three districts of Rorya, Tarime and Butiama of the Mara region, north-west Tanzania between Lake Victoria and Kenya. This area is home to several ethnic groups who are primarily engaged in agro-pastoral and fishing activities. Dog ownership is common with larger households and those having livestock tending to own more dogs [19–21]. The pilot phase included 12 wards, four from each district (three delivered the CBC-MDV strategies and one delivered the pulse). Wards are clusters of 3–4 villages; villages are divided into subvillages; the number of subvillages per village ranged from 2 to 13 in our study area. Subvillages comprise approximately 70–100 households, which are grouped into units of 10 households and headed by leaders called "mabalozi". The study wards were appreciably separated geographically and culturally. Fig 2 shows the map of the study area.

**PHASE 1 – Development of intervention(s)**

a) Initial development of components of CBC-MDV

b) Five engagement workshops with stakeholders to adapt CBC-MDV and three delivery strategies of CBC-MDV developed for piloting, accompanied with detailed implementation manual

Three delivery strategies of CBC-MDV protocol piloted in 9 wards in 3 districts in the Mara region of Tanzania

**PHASE 2 – Feasibility study of intervention(s)**

Feasibility assessed using mixed methods: observation and reviews/ audits of development, implementation processes, advertising and delivery of CBC-MDV

Data analyzed to assess: fidelity and reasons for variation; efforts by strategy delivery teams; potential effectiveness in achieving vaccination coverage and aspects of CBC-MDV needing optimization

**Fig 1. Flow of development and optimization process of the community-based continuous mass dog vaccination approach prior to full-scale evaluation.**

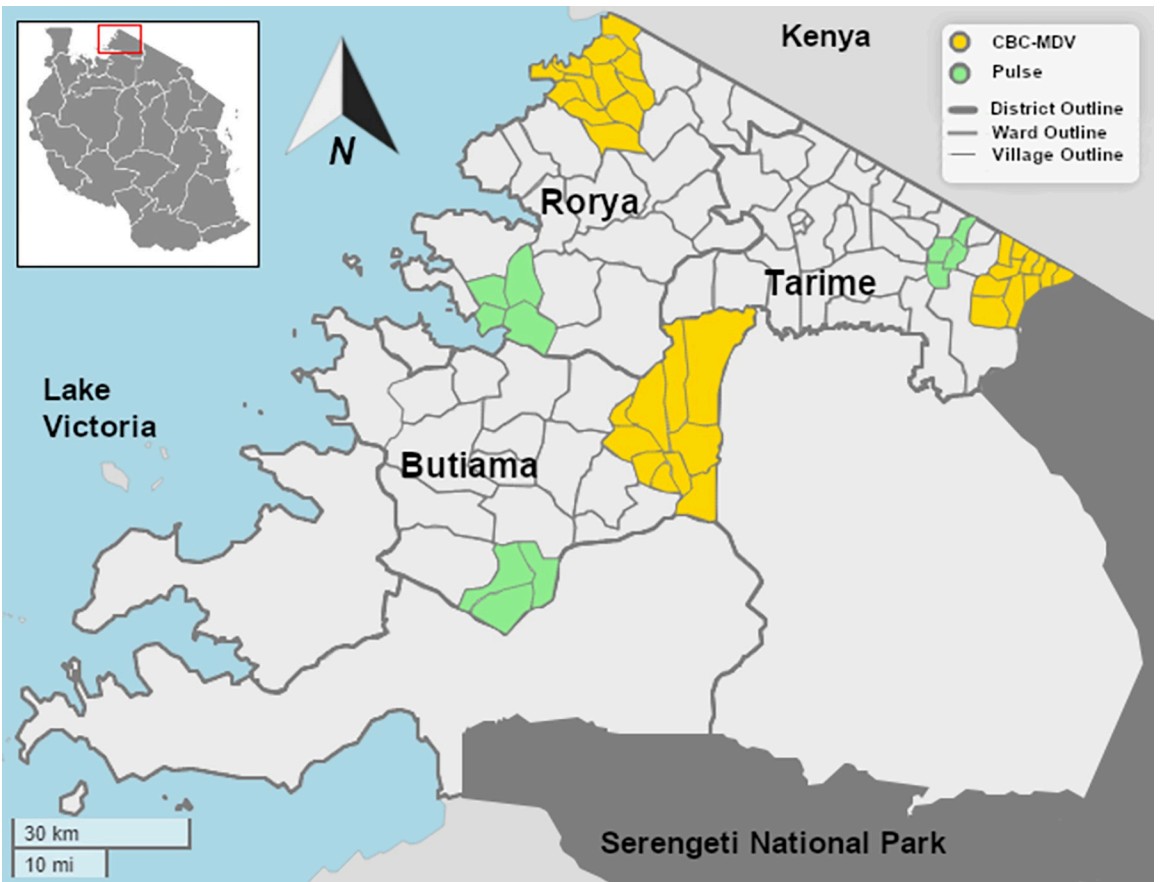

**Fig 2. Map of the Mara region showing pilot districts, wards and villages.** The surface areas of the 35 villages in the CBC-MDV study (split across the 3 wards) averaged 25.1km$^2$, ranging from 4.6–134km$^2$. The shapefiles are publicly available from the Tanzania Bureau of Statistics - https://www.nbs.go.tz/index.php/en/census-surveys/population-and-housing-census/173-2012-phc-shapefiles-level-three.

Evidence on barriers to a centralized, team-delivered dog vaccination approach (as laid out in the introduction), the feasibility of storing the Nobivac Rabies Vaccine in locally made passive cooling devices [14,15] and the ability of community-based persons to vaccinate dogs [22] provided the context for developing initial components of CBC-MDV.

The initial design was discussed with potential stakeholders in the Mara region (where a large-scale randomized controlled trial (RCT) is proposed to take place following on from this pilot study) and subsequently with national level veterinary officials and international experts, with workshops taking place between May 2018 and May 2019. Table 1 describes the stakeholder groups involved and aim of each workshop.

The first author participated in and made notes (11 observation days) of all the workshops, and documented stakeholders' opinions and concerns of CBC-MDV, specifically: how vaccines will be stored outside of the cold chain system in wards using locally made passive cooling devices, the level of training required to vaccinate dogs, local involvement in implementation and roles at district, ward and village levels. The research team met after each workshop to revise the components of CBC-MDV.

Following the final workshop, the research team developed a theory of change model and a manual to guide implementers (district livestock field officers, ward-based rabies coordinators–RCs and village-based one-health champions–OHCs) in delivering the CBC-MDV

**Table 1. Stakeholder group, purpose and date of engagement workshops.**

| | Stakeholder Group | Purpose of Workshop | Dates; Venue |
|---|---|---|---|
| 1 | National Level Veterinary Officials, Mara Regional Medical and Veterinary Officers, District Medical and Veterinary Officers, Nurses and Livestock Field Officers plus research staff | To introduce national veterinary officials and Mara region stakeholders to potential CBC-MDV strategies | 23–26 May, 2018; Mugumu–Serengeti |
| 2 | Veterinary technical staff from Ministry of Livestock and Fisheries Development, community health specialist from World Health Organization–Tanzania country office, the Mara Regional Medical Officer, representatives from Ministry of Health and Tanzanian One Health Coordination Unit plus research staff. | To share evidence for the safety of use of locally made passive cooling devices to store vaccines & non-animal health professionals to vaccinate dogs and to demonstrate that the research evidence was strong enough for local use. | 17–18 July, 2018; The Prime Minister's Office–Dar es Salaam |
| 3 | Three Rabies Researchers from Global Animal Health–Tanzania, Director of Veterinary Services and Registrar of Tanzanian Veterinary Council | To provide the outcome of Workshop 2, and to share evidence of use of locally made passive cooling devices to store vaccines & non-animal health professionals to vaccinate dogs | 17th November, 2018; Office of Director of Veterinary Services–Dodoma |
| 4 | Researchers from Washington State University (5), University of Glasgow (5), Global Animal Health–Tanzania (6), Director of Veterinary Services, Chairman and Registrar of Tanzania Veterinary Council, President of Tanzania Veterinary Association, representatives from Ministry of Health and One Health Coordination Unit | To finalize design of CBC-MDV for the pilot study, define roles of district, ward and village level implementers and to launch the research project | 22nd– 23rd Mar, 2019; Arusha. |
| 5 | Mara Regional Commissioner and Administrative Secretary, Researchers from Global Animal Health–Tanzania (6), Director of Veterinary Services, Chairman and Registrar of Tanzania Veterinary Council, President of Tanzania Veterinary Association, Mara Regional Medical and Veterinary Officers, District Medical and Veterinary Officers, Nurses and Livestock Field Officers | To bring the research team and human and animal health staff of the Mara region together, to outline logistical needs for implementing CBC-MDV and to declare the research a learning project to inform national mass dog vaccination strategies for Tanzania | 7th– 8th May, 2019; Office of the Mara Regional Commissioner |

components. To identify the most efficient approach to delivering the components, three delivery strategies of CBC-MDV were designed to be piloted.

## Phase 2: Feasibility of delivering CBC-MDV, potential effectiveness and lessons learned

The three delivery strategies of CBC-MDV were piloted over a 12-month period and evaluated using mixed methods and the outcomes compared to that of the pulse (annual team-delivered) approach which was also undertaken as part of the pilot study. Table 2 summarizes which methods were used to assess the feasibility and potential effectiveness of the delivery strategies as well as to formulate lessons learned.

**Table 2. Summary of research methods used to assess the feasibility of delivering community based continuous mass dog vaccination (CBC-MDV), potential effectiveness and formulate lessons learnt.**

| Aspect of CBC-MDV delivery assessed | Method | Data |
|---|---|---|
| Feasibility of delivery<br>  i) Fidelity to protocol<br>  ii) Reasons for resultant variation in the delivery of CBC-MDV<br>  iii) Efforts required to deliver each strategy | Observation of advertising of vaccination clinics and delivery of CBC-MDV components to assess which were delivered as intended or varied<br>Interviews with those responsible for aspects of the delivery of CBC-MDV to audit the implementation process and to capture what was delivered and reasons for variation | 36 days of observation (6/55 advertising days, 30/235 delivery days)<br>All 47 implementers at month 1 and repeated at month 6 |
| Potential effectiveness<br>  i) Vaccination coverage of the CBC-MDV delivery strategies compared with pulsed delivery | Household surveys | 1,386 and 1,445 households from 47 villages surveyed at month 1 and 11 respectively |
| Lessons for optimization and replication | Feedback and appraisal meetings of the research team examining the delivery processes and exploring feasible and effective alternative approaches | 24 fortnightly meetings; from July 2019 to June 2020 |

**Assessing fidelity, variation and effort.** To assess the fidelity of the implementation process during phase 2 and the reasons for variation in delivering CBC-MDV, we conducted observations on advertising campaigns (6/55 days) and delivery of vaccination activities (30/235 days) noting whether implementers delivered components of CBC-MDV as planned and factors responsible for variation.

We audited delivery of CBC-MDV using semi-structured interviews with implementers (one with each of the 47 implementers) about aspects of delivery, record review, inspection of how vaccines were managed at district veterinary offices and wards, and installation and maintenance of locally made passive cooling devices and their temperature loggers within wards. Notes were taken on which components of CBC-MDV were delivered as planned and on potential reasons for variation. The audits were carried out early in the delivery of CBC-MDV at month 1 and repeated at month 6.

We used observation and audit data to assess and compare efforts required for each of the CBC-MDV strategies and the fidelity of their delivery.

**Assessing potential effectiveness.** When dogs were vaccinated owners were given a vaccination certificate and dogs were microchipped. To assess how the strategies performed with respect to vaccination coverage, random samples of households (Table 2) were surveyed in each village, scanning dogs for a microchip and inspecting vaccination certificates. If neither the dog nor the certificate could be found, we asked household members whether their dog(s) had been vaccinated. The surveys were conducted at month 1 and 11 after roll out of CBC-MDV. Detailed reports of outcome measurement are presented in an outcome evaluation paper [23] and are summarized in this manuscript to provide informative context to the process evaluation.

**Lessons for optimization and replication.** To optimize CBC-MDV, the research team reviewed the observation and audit data on the delivery process through fortnightly feedback and appraisal meetings to identify components of CBC-MDV that were not working and designed alternative approaches. The team also identified best practices by implementers and components of CBC-MDV that were context sensitive. The first author participated in these meetings and made detailed reports.

## Data analysis

**Fidelity, variation and effort.** To assess the extent to which the components were delivered as intended, field notes from observations of advertising and from the audits of the implementation process were read and summarized as either 'delivered as planned', 'delivery modified', 'not delivered as planned' or 'delivered in excess of what was planned'. To assess the reasons for variation from what was planned, qualitative notes from observation of the advertising process and audits were thematically analyzed as follows. The first author developed the initial coding frame using a combination of deductive and inductive approaches [24,25]. Two authors independently applied the coding frame to a sample of the data (2 observation and 1 audit notes), and the coding frame was discussed and amended over three iterations. The first author then applied the coding frame to the whole data set. The main themes were: community engagement, estimation of dog population, advertising of campaigns, starting and closing time of vaccination clinics, delivery of continuous vaccination and choice of approaches for clinics. The coded texts were used in complementing, expanding and elaborating on understanding of the manner in which CBC-MDV was delivered and factors that influenced feasibility of delivering the different components. Qualitative data analysis was done using QSR NVIVO version 12.5.0 (NVivo qualitative data analysis software; QSR International Pty Ltd. Version 12, 2018).

To assess the effort that was required to implement each of the three CBC-MDV strategies data were collected on the number of times and hours spent advertising, and number of campaigns delivered. These data were examined to determine whether the efforts varied by strategy with bar charts plotted in Excel version 16.

**Assessing potential effectiveness.**   Vaccination coverage achieved by each delivery strategy was calculated as the proportion of the dog population surveyed that had either i) a microchip, ii) a vaccination certificate or iii) owner recall that the dog had been vaccinated. We summarized the coverage estimates at month 1 and month 11, annual averages achieved by each CBC-MDV strategy and the pulse strategy.

**Lessons for optimization and replication.**   To ensure successful replication of CBC-MDV in other settings, the research team, through the appraisal meetings, identified components of CBC-MDV that were appreciably influenced by contextual factors. Reference was made to the literature on how certain barriers to implementing community-based interventions were overcome and considered in optimizing CBC-MDV. Based on the conclusions reached by the research team, alternative approaches were designed for the CBC-MDV components that were not working as planned. Best practices among implementers were identified and incorporated into the CBC-MDV design for implementation in the full-scale trial planned for the Mara region.

## Results

### Phase 1: Development of CBC-MDV intervention

Table 3 summarizes the essential components of CBC-MDV, the rationale for their inclusion, the views on each component expressed by stakeholders during meetings and adaptations made to the design of the components to address concerns. The detailed components of each ingredient are outlined in S1 Table. The development process of CBC-MDV was iterative and participation in the workshops was multisectoral and included participants who both work in either the public health or animal health sector and are members of local communities, but did not specifically include community leaders/ decision-makers.

### The strategies of CBC-MDV tested

Stakeholders determined that the essential components of CBC-MDV could be delivered slightly differently and used the pilot (phase 2) to assess the three forms of delivery (Table 4), each of which included the essential components. A ward from each district was allocated to each of the three CBC-MDV delivery strategies. An additional ward from each district was then allocated to the pulse (once annual) strategy. The CBC-MDV and pulse campaigns were carried out over the same time period.

Fig 3 presents the logic model agreed between research team members and the stakeholder groups as to how CBC-MDV in general is expected to work.

### Phase 2: Assessment of feasibility and potential effectiveness

**Fidelity and reasons for variation.**   S1 Table presents an expanded form of the essential (45) components of CBC-MDV and summary analysis of fidelity of delivery: 20 components (44%) were delivered as planned, 14 (31%) were not delivered at all, nine (20%) were modified and two (5%) were delivered in excess of what was planned. The components were broadly categorized into eight groups (as detailed in Table 3) and their fidelity described as follows:

*i. Local delivery of CBC-MDV to be led by district level veterinary authorities to foster buy-in*: Of the four components relating to district veterinary authority roles, two were modified in

**Table 3. Essential components of CBC-MDV and responses to stakeholder concerns.**

| Essential ingredient | Rationale | Stakeholder views | Adaptation |
|---|---|---|---|
| i. Local delivery of CBC-MDV to be led by district level veterinary authorities | A new service is more likely to be adopted and sustained if it has buy-in and fits within existing systems | Stakeholders agreed district level veterinary authorities should lead implementation and suggested specific adaptations | Each district would have a district livestock field officer or a district veterinary officer who should oversee the delivery |
| ii. Involvement of village level leadership in roll out of CBC-MDV | Support from village leadership is essential for high dog owner participation and local support for sustainability | Stakeholders expected village leaders to ensure members send their dogs for vaccination | Village leadership should enforce local laws to ensure community members vaccinate their dogs |
| iii. Use of village-based people, trained prior to implementation and called One Health Champions (OHCs), to support ward-level livestock field officers to carry out vaccination activities | Local knowledge will facilitate organization and greater reach; employment of local people also provides key additional human resource | Because vaccination is professionally regulated within Tanzanian law, stakeholders would not allow people without an animal health certificate to vaccinate dogs | Each ward would have a ward-based livestock field officer Trained village-based persons (OHCs) to be allowed to register dogs and issue certificates A village-based assistant could be employed as well |
| iv. Widespread communication at village level about CBC-MDV and advertising of campaigns using multiple forms of communication and venues | Widespread communication would be essential to achieve high coverage/ reach | Use of village-based OHCs would facilitate local mobilization | Each village will have an OHC who will coordinate dog vaccination activities in the village |
| v. Use of locally made passive cooling devices to store rabies vaccine in wards | Local storage will improve operationalization of continuous dog vaccination by reducing time and travel costs thus improving access | Stakeholders agreed to storage of vaccines in locally made passive cooling devices | Livestock field officers should ensure conducive places are prepared for installation of cooling devices and their temperature monitors |
| vi. A continuous approach to MDV activities which will be delivered on a quarterly basis and is also available on demand by dog owners all year round | All year-round access to dog vaccination will support maintaining sufficient coverage necessary to interrupt transmission | Stakeholders agreed livestock field officers can devote time to organizing four rounds of vaccination campaigns in a year and to delivering vaccination on demand | Livestock field officers should collaborate with OHCs to identify dogs that missed previous rounds of quarterly vaccination |
| vii. Delivery of dog vaccination must be free of charge to the dog owners | Fees have been documented to discourage owner participation | Stakeholders agreed that vaccination of dogs and cats on this project will be free of charge to owners | To assist with the cost of implementation, local government authorities of Mara region agree to contribute US$2,000 annually to dog vaccination |
| viii. Monitoring of and feedback on vaccination coverage among research team members, district veterinary authorities, vaccinators and communities | Frequent feedback among implementers and district authorities will enable local actions to maximize CBC-MDV activities | Stakeholders agreed to monitor processes and outcomes through a joint steering committee and to reporting via the district veterinary offices | Livestock field officers must submit weekly reports to district office and research team, and provide feedback to communities |

**Table 4. Strategies for delivering components of CBC-MDV in the pilot study.**

| Strategy | Frequency | Rationale |
|---|---|---|
| One: Village level temporal static point clinics in month 1 for all villages in the ward | Campaigns repeated at months 3, 6, and 9 using either the same approach or house-to-house, plus on-demand vaccination, i.e. responding to alerts from owners of dogs needing vaccination | Within three months enough puppies and new dogs would have arrived in villages in manageable numbers for efficient vaccination |
| Two: Subvillage level temporal static point clinics in month 1 for all villages in the ward | Campaigns repeated at months 3, 6, and 9 using either the same approach or house-to-house, plus on-demand. | Bringing clinic centers closer to more people should increase owner participation |
| Three: Implementers will deliver mass dog vaccinations using whichever of the above methods they consider to be best | Continuous quarterly campaigns (at months 1, 3, 6, and 9) | Discretion to implementers and their knowledge of local terrain and context will influence their choices of vaccination approach and improve performance |

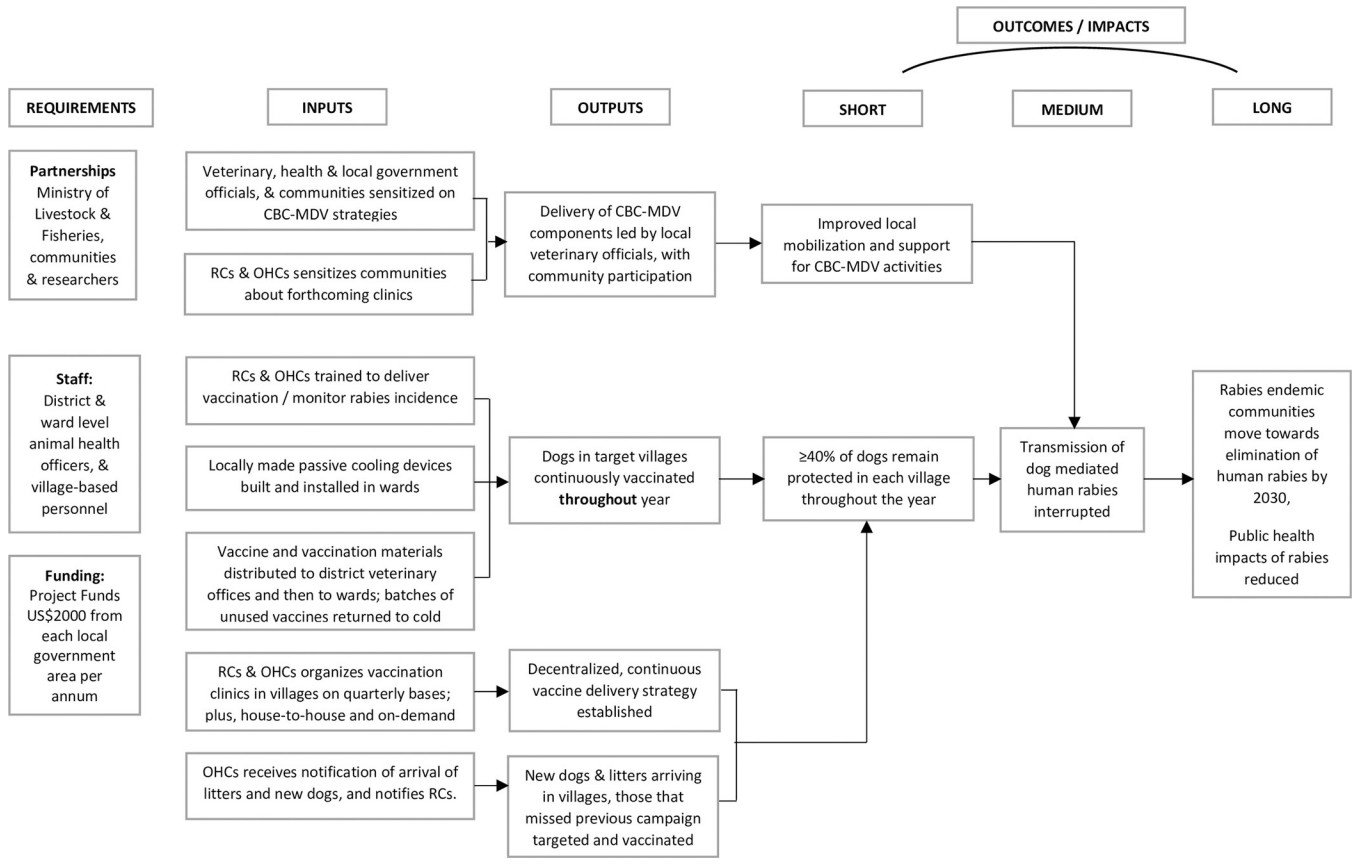

**Fig 3. Logic model of the delivery and impact mechanisms of CBC-MDV components.**

delivery. To foster community acceptance of the one health champions and rabies coordinators, the district livestock field officers were to write letters to introduce them to their villages. All the district officers wrote letters after the training workshop. The district officers took stocks of vaccines received from the research project and distributed them to wards as planned. However, vaccines returned from two wards to district offices were not labeled and stored as planned. The district officials reported only supervising and monitoring campaigns as part of routine district veterinary functions. They cited lack of vehicle and fuel as key challenges to supervision. All the RCs reported they were not supervised by district officials as planned.

*ii. Involvement of village level leadership in roll out of CBC-MDV to foster owner participation and local support*: There were five components of CBC-MDV to be implemented to bring community leadership on-board with delivery of dog vaccination. Of these, four were modified or partly delivered as planned and one was not delivered. Of 35 OHCs, the majority received letters introducing them to their villages (31, 89%). However, most of them received the letters just a few days before or after the process had started and there were very few or no opportunities to introduce them at village meetings. Of the 19 (54%) introduced, 17 were introduced only in a leaders' meeting; while in the cases of those not introduced (16, 46%), the RCs or OHCs only informed ward or village executive officers about the program. Hence, most villagers did not have the opportunity to link the RCs and OHCs with the vaccination campaigns before they started.

The protocol also required RCs to discuss vaccination timetables with village leaders; only four out of nine RCs reported directly informing a community leader about their timetables. Again, OHCs were to work with 'mabalozi' to estimate the village dog population. These were partly implemented; only a few (3, 9%) OHCs reported working with 'mabalozi'; the rest either went to houses directly (19, 54%) or instead worked with subvillage chairpersons (13, 37%). The frequently cited reasons for not working with 'mabalozi' included: 'mabalozi' perceived OHCs as not belonging to their political party or did not see the project as a community agenda and hence requested money (15, 43%); *"one 'balozi' said, you went to the workshop and received big allowances and you have come to tell us to go and work"* [OHC, Implementation Audit, Strategy 1-Tarime]. Also, the concept of 'mabalozi' is not practiced uniformly across all jurisdictions (11, 31%). Other reasons were: OHCs thought they were to work instead with subvillage chairpersons (6, 17%) or they did not trust 'mabalozi' to produce accurate figures on the dog population (4, 11%).

*iii. Use of trained village-based One Health Champions to support ward-level rabies coordinators with local knowledge to carry out vaccination activities*: There were six essential ingredients relating to village-based personnel supporting delivery of CBC-MDV at village levels. Out of these six, two were delivered as planned, one was partly delivered, two were not implemented and one was implemented in excess of what was planned. To ensure that only the required number of vaccines for a round were requested, all OHCs (35, 100%) provided estimates of the village dog population to RCs for request of vaccination materials. All OHCs also advertised vaccination clinics as planned. On the other hand, only two out of 35 OHCs conducted sensitization in village meetings. The opportunities for OHCs were limited as most of the villages did not hold meetings before the start of campaigns. Over the course of the year none of the OHCs documented dogs that missed the previous rounds as planned. All OHCs supported vaccination clinics in other villages of the ward in addition to theirs, as the workload at a center is ideally for three people. Not all OHCs had cooperation from their village leadership, most of the OHCs were not persons with influential village positions.

*iv. Widespread communication at village level about CBC-MDV and advertising of campaigns using multiple forms of communication and venues to promote high reach*: Advertising of campaigns was largely carried out as planned. Of three components relating to advertising, one was delivered as planned, one modified and one delivered in excess of what was planned. All OHCs (35, 100%) delivered the complete contents of the adverts as designed, which included: date, time, location of clinic, specified animals to be vaccinated as dogs and cats, and vaccination being free-of-charge, using megaphones and posters at vantage points. However, instead of the night before, advertising started two to three days before, occasioned by perceived workload (nature of settlement and size of villages–need to cover long distances). Out of a total of 55 adverts of the first round of campaigns, only 24 (44%) were carried out in the evenings; the rest were carried out in mornings (20, 36%) or afternoons (11, 20%) in variation with the protocol.

*v. Use of locally made passive cooling devices to store rabies vaccine in wards to support provision of continuous vaccination*: To ensure vaccines do not remain outside of the cold chain for more than six months, eight CBC-MDV components were to help to deliver the vaccines to wards in batches. Six out of these were implemented as planned including: coordinated requests and transport systems, basing requests on ward dog population, returning unused vaccines after six months, installation of cooling pots away from sunlight and monitoring daily temperature in pots. However, labeling of unused vaccines was not carried out as planned; only two out of nine RCs reported having ever returned unused vaccines to the district office and these were given to wards which were not part of the studies for use. Four out of nine pots

were not in full use because they developed cracks and leaked when water was added to the cooling sand layer.

The prescribed waste management plans were partially implemented. The different kinds of waste were mostly separated during vaccination clinics (7/9), but instead of sending metallic and biohazard wastes to district offices or nearest health centers for incineration, most teams burnt everything at the location of clinics (6/9), indicating it was safe to do so.

*vi. A continuous approach to MDV activities; quarterly basis and available on demand by dog owners all year round thereby providing continuous access to dog vaccine*: Of the five components of CBC-MDV targeted at supporting provision of continuous dog vaccination, two were implemented as planned, one was modified and two were not implemented as planned. The CBC-MDV protocol prescribed that each strategy team conducts four rounds of campaigns in a year. However, only three out of the nine teams conducted four rounds of campaigns. The frequently cited reasons for variation in vaccination schedules included: farming/rainy seasons, national activities such as elections, counting of poor households and mass animal vaccination campaigns (in which some RCs participated), social events such as cattle auction days, funerals, puberty rites celebrations and school cycles, with campaigns more patronized on weekends during school terms. For example, some dog owners indicated that during the farming season, either they or their dogs were required in the farms during the day time to guard against monkeys destroying their crops. It was also noted in one district that campaigns were halted during the month-long puberty rites celebrations.

The activity of finding unvaccinated dogs that missed previous rounds was not implemented as planned. The implementers cited that this activity was labor-intensive and not feasible in the absence of an existing village register of dogs. To ensure dog owners have easy and continuous access to vaccinators, the protocol prescribed that OHCs give their mobile numbers out during first round of campaigns. None of OHCs reported giving their numbers out directly as planned (0, 100%) but most (32, 91%) wrote them on the 5–10 posters per village they pasted. The research team observed giving numbers out was practically difficult to do during advertising or vaccination given how busy they were at the centers. However, more than half of OHCs (20, 57%) reported having received calls from dog owners to visit their homes to vaccinate their dogs.

*vii. Delivery of free dog vaccination clinics using suitable approaches to encourage owner participation*: Out of the eight components related to organizing vaccination clinics, five were implemented as planned, one was modified and two were not implemented as planned. The CBC-MDV protocol prescribed that vaccination should take place between 08:00–14:00; in practice clinics started as early as 07:00 and as late as 12:00; and closed as early as 11:00 and as late as 18:00. The length of clinics was dependent on turnout at centers. House-to-house campaigns took longer where houses were further apart. The starting time for clinics depended on when farmers had returned home, whether RCs had to perform other duties on the same day (e.g., having to inspect meat) before clinics or whether RCs had to attend to personal business. Vaccinators also cited that microchipping dogs (during which a number of dogs struggled) and entering data into the digital data collection device was time-consuming.

To ensure safe vaccination of dogs by reducing dog aggression, the implementation manual prescribes separation of registration and inoculation points with at least a 20-meter distance and muzzling of potentially aggressive dogs. However, none of the vaccination teams (0/9) implemented these. Dog aggression was associated with poor dog handling techniques by vaccinators. Dog aggression was observed to increase the time-per-dog vaccinated and on rare occasions resulted in injury, especially of dog owners.

Muzzles were not used out of fear of being bitten or the muzzles could tear in the process. One rabies coordinator said: "*is too difficult to use muzzles, dogs are too fierce to use it on them,*

*it will get loose, we are afraid, we use the Y-stick*" [RC, Implementation Audit, Strategy 2-Tarime]. Others recommended muzzles of three different sizes, whilst others perceived use of muzzles as time consuming. Consequently, implementers in Butiama and Rorya Districts restrained aggressive dogs by tying the rope or chain on the neck of dogs closely to a tree, and holding the hind legs firmly whilst inoculating the dog. While those in Tarime District used a 'Y-stick' to pin down the dog at the neck region with the help of the rope or chain.

The vaccination teams varied the delivery strategies that were prescribed for them, citing the following reasons: villagers saying it was difficult to bring dogs over long distance to centers, large dog populations in their villages, and their own perception of which strategy was likely to reach more dogs. Remarks by implementers indicated they thought subvillage level temporal static point clinics was the most effective approach, with the following quotes exemplifying this, "*subvillage level is very good at reaching more dogs*" [RC, Implementation Audit, Strategy 1-Rorya]; "*the Strategy (subvillage level temporal static point approach) is good because we had time to educate the dog owners*" [RCs, Implementation Audit, Strategy 2-Butiama]; "*I think Strategy 3 is good, it covers a lot of places because we use sub-village level (temporal static point approach), house to house and on demand*" [RC, Implementation Audit, Strategy 3-Tarime].

*viii. Monitoring and feedback on vaccination coverage among stakeholders to promote collaborative local action*: Of the six components relating to monitoring, reporting and providing feedback on CBC-MDV, only two (RCs reporting on dogs vaccinated and daily temperature recording of the low-tech cooling devices, and rabies cases) were delivered as planned. Supervision of campaigns by district veterinary officers was not carried out; the district veterinary officers cited lack of transportation to carry out this task and they expected per diem payment while supervising. OHCs also did not provide weekly tallies of dogs needing vaccination, they considered the weekly submission of tallies too frequent to allow for completion. Communities' self-monitoring of the campaigns and reporting back to the research team and the district veterinary office were also not carried out, largely due to weak community involvement in the design and delivery of CBC-MDV.

## Comparing efforts made at delivering CBC-MDV components by strategy teams

*Involvement of village level leadership in roll out of CBC-MDV*: The strategy teams delivered components relating to involving village leaders with varied degrees of fidelity. For example, very few OHCs discussed their timetables with a village leader to get their approval and support (0/12 for Strategy 1, 2/13 for Strategy 2 and 2/10 for Strategy 3). OHCs largely did not work with "mabalozi" to estimate the dog population in their ward: Strategy 1 (3/12), Strategy 2 (0/13) and Strategy 3 (0/10). Further information about how the delivery of the additional components were delivered is provided in S2 Table.

All components relating to use of trained village-based OHCs to support vaccination were delivered as planned by all strategies, except sensitization of villagers about campaigns at village meetings. All strategies delivered advertising components as required, but the effort put into the advertising differed: The number of times and hours per village advertised in the first round, and total number of days of vaccination per village were all lowest in Strategy 1 and highest in Strategy 3 respectively (Fig 4).

The vaccinators reported that having to walk for a long distance or personally pay for travel by motorbike created challenges to advertising. On average, each OHC had to cover an area of 25.1km$^2$ during advertising.

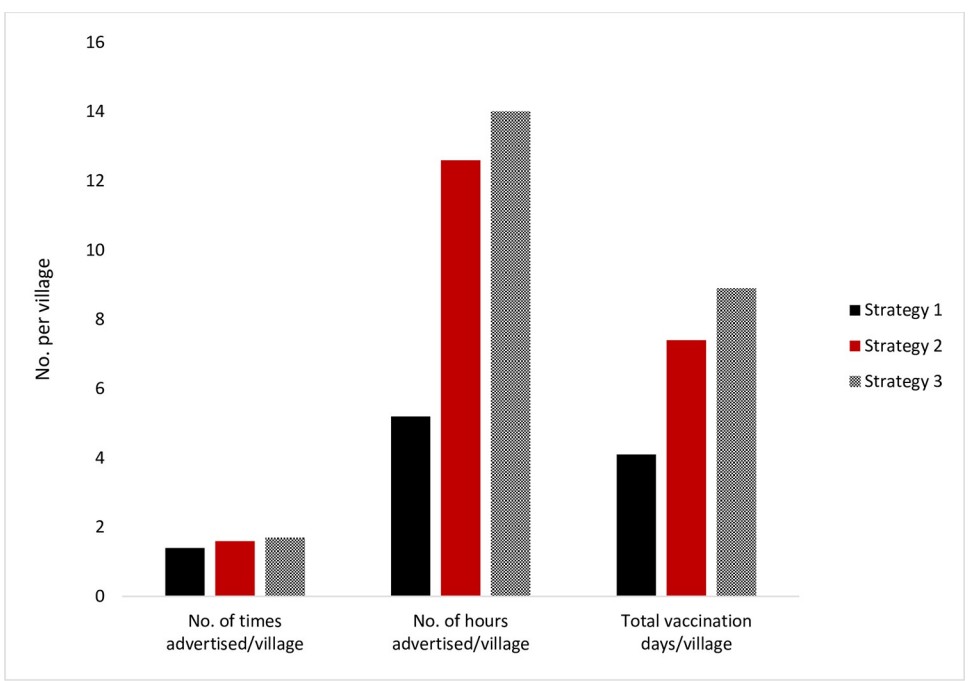

**Fig 4. Effort at advertising and delivering vaccination campaigns by strategy (totals for all three team per strategy).**

For all strategies, the number of days of campaign activities reduced drastically after the first round. Over the one-year period, the three strategies together used 237 days on campaigns: Strategy 1 (49, 21%), Strategy 2 (95, 40%) and Strategy 3 (91, 39%). The majority of days (189 days, 80%) were spent during the first two rounds (Fig 5).

None of the strategy teams went house to house to find dogs that missed central point clinics. They spent varying number of days responding to on-demand vaccination by dog owners and in organizing quarterly campaigns (S2 Table).

*Waste management after vaccination clinic*: All teams installed and managed vaccine batches as planned. However, there was discrepancy with regards to how used needles and microchip units were disposed. Some teams either incinerated or disposed of these items in pit toilets: Strategy 1 (2/3 teams), Strategy 2 (2/3 teams) and Strategy 3 (1/3 teams), whilst the rest of the teams burnt all waste at vaccination centers (S2 Table).

*Delivery of free dog vaccination clinics using suitable approaches*: none of the Strategy teams implemented separating registration and inoculation centers with a distance of at least 20 meters and muzzling of potentially aggressive dogs as planned. The Strategy teams partly followed the CBC-MDV manual in selecting approaches to deliver dog vaccination: All Strategy 3 wards opted for subvillage level temporal static point approach, the same approach as was prescribed for use in Strategy 2 wards in round 1 (6/6). In round 2, two of the Strategy 3 wards avoided the lengthy campaign days that come with subvillage level temporal static point approach by deciding to use village level temporal static point. A remark by an RC exemplifies this: "*it (subvillage level temporal static point approach) took long*" [RC, Implementation Audit, Strategy 3-Rorya]. Conversely, two out of the three Strategy 1 teams switched from village level in round 1 to subvillage level temporal static point approach in round 2. The reason given for this switch was that many dogs remained unvaccinated after the round 1 village level temporal static point clinics and so they decided to instead employ a subvillage level temporal static point approach to reach more dogs. All teams employed some house-to-house and on-demand

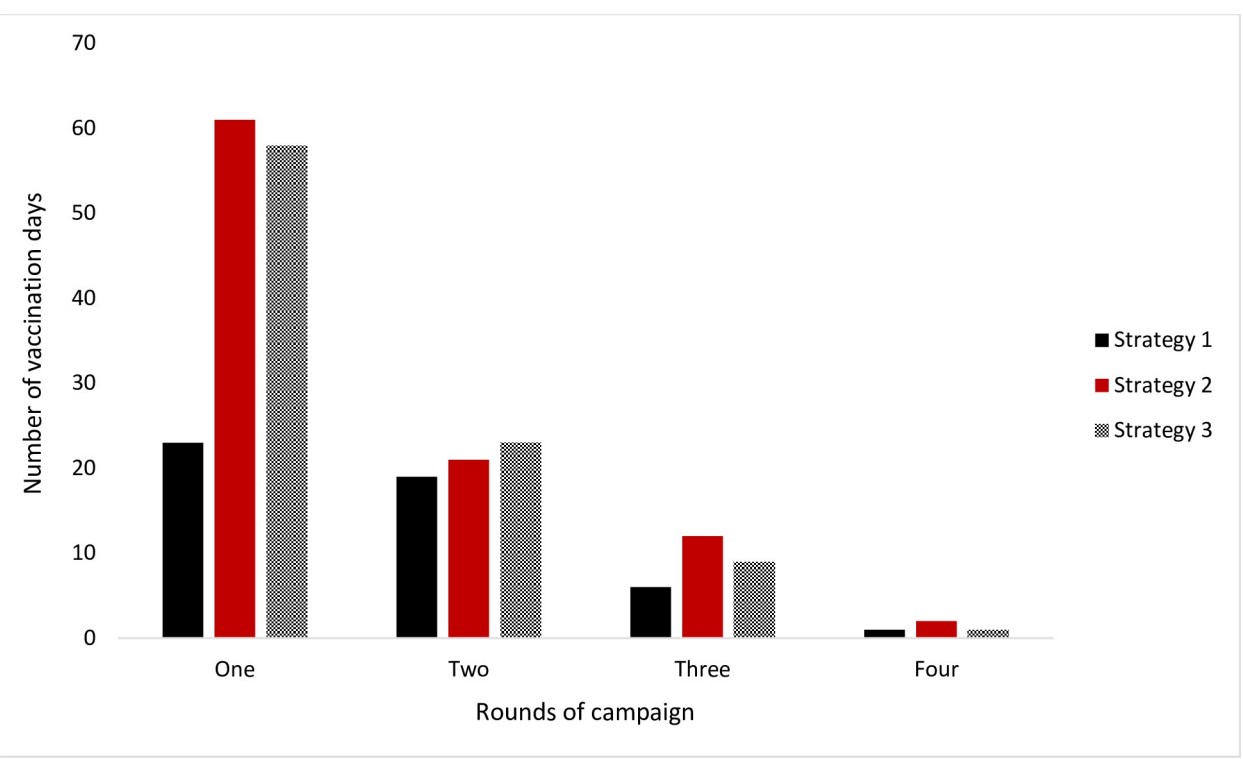

**Fig 5. Number of days implementers conducted vaccination activities during each round (totals for all three team per strategy).**

(9/9) approaches. Subvillages were combined for single clinics when implementers considered them to be smaller in size, had smaller dog populations or were closer to each other (S3 Table).

Overall, subvillage level temporal static point and on-demand approaches were the most (173 occasions) and least-used (20 occasions) respectively (Fig 6).

### Potential effectiveness of the CBC-MDV strategies

To interrupt rabies transmission requires sustaining vaccination coverage above the critical vaccination threshold (approximately 40%). Coverage estimations at month 1 and 11 showed all continuous strategies did sustain coverage above this level, whilst the pulsed approach did not achieve the ≥70% target (Table 5). Coverage at month 11 was slightly lower in Strategy 1 and 3 and slightly increased in Strategy 2, but none were significantly different (Table 5). Strategy 3, which recorded the highest work inputs in terms of advertising and vaccination days, also recorded slightly higher annual average vaccination coverage: Strategy 1, 2 & 3 (61.43%, 62.93% & 63.46%), respectively (Table 5).

### Optimization of CBC-MDV for replication in the full-scale trial and dissemination in other contexts

Table 6 details optimization of some components of CBC-MDV for replication in the full-scale trial and lessons for dissemination in other contexts.

### Discussion

This paper provides formative insights into the development, feasibility, potential effectiveness and optimization of a community-based, continuous mass dog vaccination approach

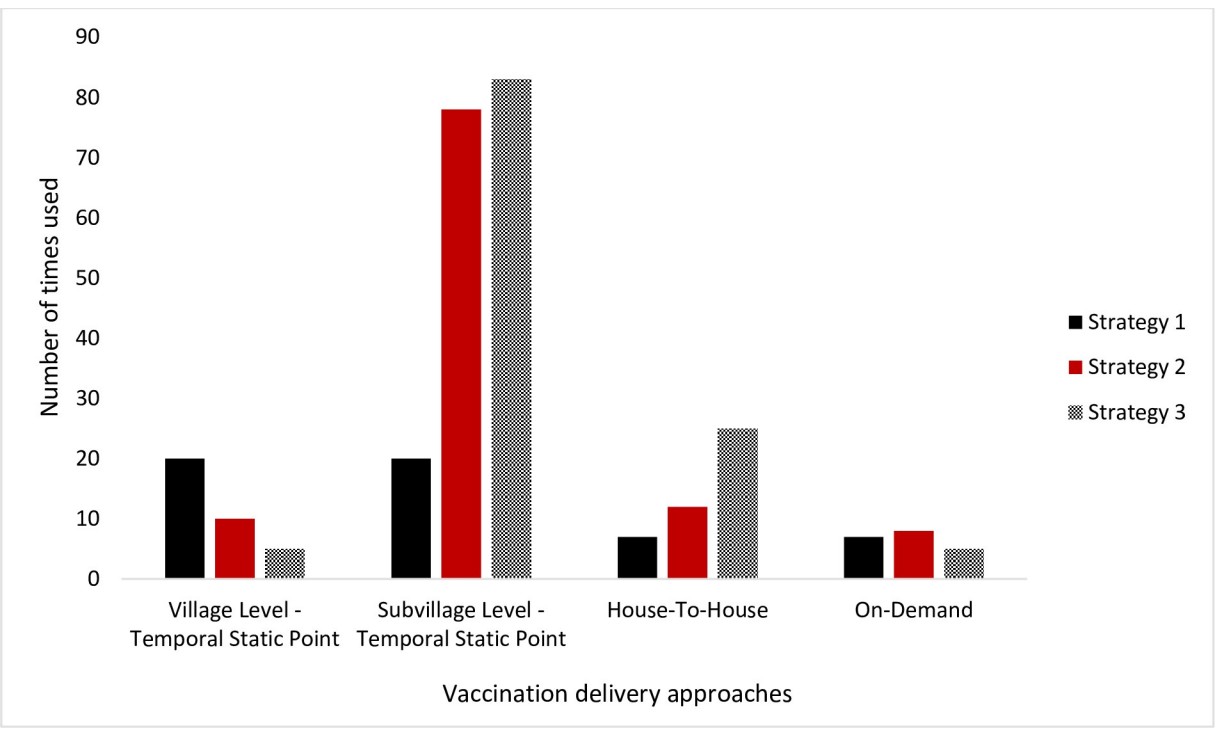

**Fig 6. Use of vaccination delivery approaches by strategy team (totals for all three team per strategy).**

(CBC-MDV). The key findings were: *i*. The development process of CBC-MDV was iterative and involved stakeholders from multiple sectors but did not include direct involvement of representatives from the target villages; *ii*. It was feasible to deliver about half of CBC-MDV components as planned (about 50% fidelity to implementation manual); *iii*. Variation of delivery from what the implementation manual prescribed was because of factors inherent in the design of the CBC-MDV strategies, implementers' understanding and appreciation of the CBC-MDV components and moderating effects of contextual (sociocultural, economic, political and environmental) elements such as elections, mass cattle vaccination campaigns, livestock auction days, funerals, puberty rites celebrations and school cycles; *iv*. All the delivery strategies of CBC-MDV sustained vaccination coverage above the critical threshold (approximately 40%), whilst the pulse (once annual) strategy failed to achieve the required ≥70% vaccination coverage; and *v*. Because of the variation from what the implementation manual prescribed, a number of CBC-MDV components needed optimization prior to replication in the planned full-scale trial.

The absence of community involvement in the design stage of CBC-MDV and weak community sensitization at roll out likely explains why some village leaders perceived the project

**Table 5. Vaccination coverage achieved by the delivery strategies at month 1 and 11.**

| Vaccination coverage achieved by delivery strategies | | | |
| --- | --- | --- | --- |
| Strategies Arms | Month– 1 (%) | Month– 11 (%) | Annual Averages (%) |
| Pulse | 35.86 | 32.10 | 33.98 |
| Strategy 1 | 65.07 | 57.78 | 61.43 |
| Strategy 2 | 60.97 | 64.88 | 62.93 |
| Strategy 3 | 68.00 | 58.91 | 63.46 |

**Table 6. How CBC-MDV can be optimized for replication in the full-scale trial and dissemination in other contexts.**

| Optimization of MDV-CBC design for the full RCT in response to delivery challenges | |
|---|---|
| Delivery challenge | How the delivery has been modified |
| Dog aggression | Feasibility of a facial recognition application is being tested in the RCT as a means of identifying vaccinated dogs instead of microchipping to avoid microchipping needles irritating dogs and making them aggressive. Vaccinators will be trained on dog behavior and handling techniques |
| Microchipping was time-consuming | Facial recognition application is relatively faster |
| OHCs not receiving maximum cooperation from "mabalozi" | Village chairpersons were selected to be OHCs, to use their authoritative positions to enhance community engagement and sensitization, potentially leading to improved community acceptance, support and participation in vaccination campaigns |
| **Further potential optimization of MDV-CBC design that can be made to overcome other delivery challenges** | |
| Dog aggression | The sequence of procedures at clinic centers can be reordered; where painless procedures such as tying of collars are carried out before painful procedures such as inoculation. This potentially will avert dog aggression and bites of owners while tying collars |
| Delivery of CBC-MDV components being affected by community level environmental, economic and sociocultural factors such as elections, mass animal vaccination campaigns, cattle auction days, funerals, puberty rites celebrations and school cycles | Inclusion of community leaders in planning of CBC-MDV could lead to integration of CBC-MDV into village annual calendars (highly revered and largely adhered to), potentially improve tailoring of delivery to local events. Village authorities will be more inclined to earmark resources towards CBC-MDV implementation: transport and launch allowances for vaccinators, volunteers to assist clinics, enforcing dog vaccination and community self-monitoring of campaigns |
| Identifying dogs that missed previous campaigns being labor-intensive | Campaigns can begin with a census of the entire village dog population linked to households, and will be ticked as dogs are vaccinated. Thus, dogs that missed a round of vaccination and where they live can easily be identified and targeted. This potentially will facilitate effective logistics planning, accurate coverage estimation and delivery of continuous vaccination |
| Implementers finding it challenging to give their telephone numbers out during vaccination clinics | Vaccination cards can be printed with the telephone number of the RC of the ward on them. This would allow villagers ready access to vaccinators and potentially will promote on-demand/ continuous vaccination |
| RCs' routine duties and personal businesses influenced timing and frequency of vaccination schedules | Schedules composed of 3-rounds of vaccination (at the village / sub-village level) per year will be more manageable for RCs given their other duties. The campaign must include robust arrangements for on-demand to target new dogs and puppies that arrive in the village and dogs that missed previous vaccination rounds |
| OHCs having to participate in campaigns for each village of the ward | The work load at a vaccination center ideally requires three people. Hence provisions should be made to support OHCs/ volunteers to assist campaigns in other villages. |

(*Continued*)

**Table 6.** (Continued)

| Optimization of MDV-CBC design for the full RCT in response to delivery challenges | |
|---|---|
| Lack of supervision of vaccination campaigns by district veterinary officers | Frequent supervision and higher number of days spent vaccinating can be encouraged by a remuneration system that is based on performance: a portion of implementers' salaries can be paid as bonuses/ allowances upon delivery of certain indicators: e.g., for RCs—carrying out all rounds of campaigns of the year, complete & timely monthly reporting, achieving coverage above a minimum threshold at month 11, no animal rabies cases recorded in the ward; for district veterinary officers–number of verifiable supervision days, number of feedback provided to research team and or communities |

as an avenue for making money, questioned the identity of OHCs or did not fully cooperate. More effective community entry processes could have enhanced participation, strengthened collaborations between implementers and community leaders in mobilizing towards vaccination campaigns. This, potentially, could have led to increased community support and contributions to the implementation of CBC-MDV [26–30]. To enhance community involvement, project implementers must allow enough time for community entry and engagement processes to take effect before commencement of vaccination campaigns.

Globally, community participation in intervention delivery has evolved from communities as passive recipients, through communities as active participants in delivery to communities as co-designers of interventions [17,27]. The performance of the community-based personnel in the delivery of CBC-MDV components and outcomes of community-led interventions elsewhere show that communities can implement intervention such as dog vaccination campaigns if effectively engaged and supplied with logistics [27,31].

Implementation of CBC-MDV components that relate to managing vaccination logistics, organizing clinics and information recording were carried out with high fidelity. On the other hand, components aimed at ensuring that vaccination clinics proceeded smoothly such as community engagement, supervision of campaigns, separation of registration and inoculation points to minimize dog aggression were mostly omitted or implemented with low fidelity. Certain components such as finding dogs that missed previous rounds, vaccinators giving their telephone numbers out to dog owners at centers and muzzling of potentially aggressive dogs appeared practically challenging to implement. For instance, some implementers expressed fear about muzzling a dog, others indicated the muzzles were too small or could tear in the process. This is line with findings by other process evaluation studies, where implementers not having ample time to assimilate the value of intervention components, not feeling competent enough to deliver certain components or having unusable equipment resulted in low fidelity [32,33].

The challenge of dog aggression could be surmounted with use of oral vaccination approaches, either broadly or targeted at dogs that are difficult to restrain. Also, potentially, the CBC-MDV approach can be used to deliver oral rabies vaccination by communities adjacent to national parks to target wildlife (foxes, racoons, hyenas, wild dogs etc) that have frequent interactions with domestic dogs. However, studies conducted in Zimbabwe and Tunisia showed only 17% of 369 baits were chewed after 24 hours, indicating the need to distribute a large number of baits [34] and required more time to implement respectively [35]. Cost-effective evaluations of using oral bait versus mass injection must be made to inform the choice of approach.

The marked decline in the number of vaccination days with each passing round of vaccination may be an indication of implementation fatigue. RCs serve large populations (3–4 villages / ward on average) by providing many different extension services such as dipping of large herds of domestic animals, meat inspection at several different locations, animal levy collection at cattle auctions and other routine duties. It is likely that conducting four rounds of dog vaccination campaigns alone was a substantial additional workload. It is also possible that the RCs did not consider the *continuous* component of CBC-MDV very critical, and assumed that they had vaccinated sufficient dogs in Round 1 without much consideration of arrival of new dogs and puppies in villages. This is consistent with the findings of other studies, which cited staff 'burn out' as a barrier to implementing community-based interventions as intended [36,37]. How much work CBC-MDV adds to routine duties of implementers would be a useful consideration during replication and scale up of the approach.

The variation in work inputs across the different strategies was influenced primarily by the design of the respective strategies. Though this was not statistically tested due to the limited sample size, to show how the timing and amount of advertising, as well as number of days used in conducting campaigns influenced coverage, it offers insights into the differences in the coverage achieved by each strategy (Fig 4 & Table 5).

Strategy 1 required a larger effort over a shorter period of time for the implementers. However, because the vaccination activity of Strategy 1 occured at a central point of the village, for many owners this strategy likely posed a challenge of access as they will be required to travel further to reach the central point. Living far from the point of the clinic has been cited by other studies as a reason for nonparticipation in vaccination clinics[38–41].

In comparison, Strategy 2, being hosted at the subvillage level, came with a relatively lighter workload on each vaccination day for the implementers. However, with multiple subvillages for every village, it required multiple days to complete the campaign (reaching 35 consecutive days). However, subvillage level clinics are easier for the owners to attend because of their closeness. It is noteworthy, that when given the discretion to choose, all Strategy 3 teams adopted the subvillage (Strategy 2) approach even though they reported it required more time. Suggesting that, empowering the implementers to select approaches fostered a stronger sense of ownership and desire to work harder. This notion is supported by previous research where social motivation was found to enhance community participation in community level development activities [42]. The discretion also may have allowed Strategy 3 teams to be more flexible in their schedules around personal and local events.

Strategy 3 teams also recorded higher number of times and hours advertised per village and number of vaccination days per village, and this possibly explains why the annual average vaccination coverage achieved by Strategy 3 was marginally higher [23]. However, the ability to use their discretion may have caused Strategy 3 teams to relax after the first round of clinics as they accounted for 3 out of the 7 missed rounds by all strategies and could be why they recorded a lower coverage at month 11. Given the differences in the prescribed activities, it seems logical that Strategy 1 teams would need to work harder during subsequent rounds to attain similar outputs as strategies 2 and 3. Frequent supervision from district veterinary officials and oversight by community leaders could have helped to achieve higher levels of campaign activities during subsequent rounds.

Several local environmental, economic and sociocultural events also affected the feasibility of delivering the CBC-MDV components. Structural community participation in initializing and implementing the intervention could help take these events and issues into account during planning and delivery. Consequently, replication of CBC-MDV across wider contexts would benefit from tailoring campaign schedules to local environmental and social events [40,41,43,44] or calendars. In this regard, the CBC-MDV strategy would be less impacted by

local events compared to the pulse strategy. For instance, whilst both strategies can be affected by timing of local events, the impact of disruption on a single day of a pulsed campaign would likely be more significant than the impact on several days of CBC-MDV campaigns as this latter strategy provides more timepoint access than the pulse strategy.

Including communities in evaluating outcomes of CBC-MDV is likely to foster ownership and sustained efforts at delivering components. Community participation in evaluating local interventions has been gaining traction and, for example, was a key component of the community-directed treatment with ivermectin (CDTI) model introduced by the African Program for Onchocerciasis Control [17,27]. In the CDTI model, a 3-member committee selected by each village carried out community self-monitoring of mass distribution of ivermectin, thereby checking the performance of distributors and compliance of community members. In the process, challenges were identified and resolved with participation of community leaders. Lessons and strategies such as those outlined above and those generated from this study could be incorporated into CBC-MDV to ensure its successful replication.

Process evaluation has been carried out for a wide range of complex interventions, but to our knowledge, this study represents the first process evaluation of mass dog vaccination campaigns. The study revealed implementation bottlenecks in the delivery, the understanding of the impact pathways underpinning these bottlenecks and also opportunities for addressing them. These insights could be of value when designing national rabies elimination strategies.

The study is likely slightly affected by recall bias where data collection processes depended to a large extent on implementer reports. However, the use of mixed methods approach, including non-participant observations and following the intervention prospectively through the design and implementation phases provided first hand data.

## Conclusions

The development of CBC-MDV incorporated extensive stakeholder views, leading to stakeholder acceptance of the approach. Including community level decision makers/ leaders in the process will likely foster ownership among communities as well. Intervention-, implementer- and context-related factors influenced delivery of CBC-MDV components and effectiveness of the strategies at reaching more dogs. The CBC-MDV strategies sustained vaccination coverage well above the critical threshold (approximately 40%) throughout the year whilst the pulse strategy failed to achieve the required vaccination coverage of $\geq$70%. The findings are being used to optimize the CBC-MDV components for dissemination in an RCT across the entire Mara region. Overall, we conclude that improved supervision and monitoring, as well as community participation in designing, planning and implementing of the dog vaccination campaigns could result in higher fidelity and reach of the CBC-MDV strategies in a more sustainable manner.

## Supporting information

**S1 Table. Delivery of 45 components CBC-MDV compared to what was planned.**
(DOCX)

**S2 Table. Comparison of fidelity and reasons for variations in delivery of components of CBC-MDV by strategy arms.**
(DOCX)

**S3 Table. Utility of approaches and number of rounds of vaccination clinics organized by strategy arms.**
(DOCX)

## Acknowledgments

The authors would like to acknowledge the government of the United Republic of Tanzania for providing a conducive environment to implement this research. The authors wish to extend their gratitude to the host research institutes (Ifakara Health Institute, Nelson Mandela African Institution of Science and Technology) and collaborating partners; the National Institute for Medical Research, the Tanzania Commission for Science and Technology for permission to undertake this research. The authors wish to acknowledge David Anderson for making the study map. The content is solely the responsibility of the authors and does not necessarily represent the official views of the National Institutes of Health.

## Author Contributions

**Conceptualization:** Christian Tetteh Duamor, Katie Hampson, Felix Lankester, Sarah Cleaveland, Sally Wyke.

**Data curation:** Christian Tetteh Duamor, Ahmed Lugelo.

**Formal analysis:** Christian Tetteh Duamor, Ahmed Lugelo.

**Funding acquisition:** Katie Hampson, Felix Lankester, Sarah Cleaveland, Sally Wyke.

**Investigation:** Christian Tetteh Duamor, Ahmed Lugelo.

**Methodology:** Christian Tetteh Duamor, Sally Wyke.

**Project administration:** Christian Tetteh Duamor, Felix Lankester, Ahmed Lugelo.

**Resources:** Christian Tetteh Duamor.

**Software:** Christian Tetteh Duamor.

**Supervision:** Katie Hampson, Felix Lankester, Emmanuel Mpolya, Katharina Kreppel, Sarah Cleaveland, Sally Wyke.

**Validation:** Christian Tetteh Duamor, Emmanuel Mpolya.

**Visualization:** Christian Tetteh Duamor.

**Writing – original draft:** Christian Tetteh Duamor.

**Writing – review & editing:** Christian Tetteh Duamor, Katie Hampson, Felix Lankester, Katharina Kreppel, Sarah Cleaveland, Sally Wyke.

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
