## [Decision Letter · Decision Letter 0]

14 Jun 2022

Dear Mr. Duamor,

Thank you very much for submitting your manuscript "Development, feasibility and potential effectiveness of community-based continuous mass dog vaccination delivery strategies: lessons for optimization and replication" for consideration at PLOS Neglected Tropical Diseases. As with all papers reviewed by the journal, your manuscript was reviewed by members of the editorial board and by several independent reviewers. The reviewers appreciated the attention to an important topic. Based on the reviews, we are likely to accept this manuscript for publication, providing that you modify the manuscript according to the review recommendations. 

Sincerely,

David Joseph Diemert, M.D.

Associate Editor

Gregory Moseley

Deputy Editor

Reviewer's Responses to Questions

**Key Review Criteria Required for Acceptance?**

**Methods**

-Are the objectives of the study clearly articulated with a clear testable hypothesis stated?

-Is the study design appropriate to address the stated objectives?

-Is the population clearly described and appropriate for the hypothesis being tested?

-Is the sample size sufficient to ensure adequate power to address the hypothesis being tested?

-Were correct statistical analysis used to support conclusions?

-Are there concerns about ethical or regulatory requirements being met?

Reviewer #1: No new analyses required.

The methods were well conceived and implemented. The majority of the methodology was well explained. Please see the general comments below for some minor clarifications.

Reviewer #2: This paper is largely a descriptive study of a highly complex process aimed improving rabies control in Tanzania. The process was described very well and provides a template for future studies. The details about the process are sufficient to develop an improved study for this region and for translation to other parts of the world and diseases.

**Results**

-Does the analysis presented match the analysis plan?

-Are the results clearly and completely presented?

-Are the figures (Tables, Images) of sufficient quality for clarity?

Reviewer #1: The results are clear and well articulated. Please see the general comments below for some minor clarifications.

Reviewer #2: The figures are marginally useful. The legends are inadequate to allow the reader to fully interpret the results in the figure without needing to flip back through the manuscript to find relevant information.

Table 6 should be completely redesigned.

**Conclusions**

-Are the conclusions supported by the data presented?

-Are the limitations of analysis clearly described?

-Do the authors discuss how these data can be helpful to advance our understanding of the topic under study?

-Is public health relevance addressed?

Reviewer #1: The conclusions are clear and match the results presented. Clear limitations and next steps were provided.

Reviewer #2: The limitations are clearly identified, and this study has great potential to change vaccination processes in rural areas. 

This was a fantastic study with a great level of detail included in the manuscript, but was challenging to read at times. The final conclusion paragraph on lines 708-724 could be much simpler so that anyone that reads part or all of the document retains and simple take away story.

**Editorial and Data Presentation Modifications?**

Reviewer #1: General: Please check the spelling consistency between US and UK English throughout. E.g., line 350, the word “programme” (UK English) is used, while line 383, “center” (US English) is used. 

Table 3. Point v, third column: The words “in locally made” are repeated. Please correct. 

Line 361: Please change “seen” to “saw”. The grammar of the sentence can be improved by saying, “or did not see the project…”. 

Line 384: “Not all of OHCs had cooperation…” Please either delete the “of” or include “of the”.

Line 394: Mega phones – please correct to megaphones (one word). 

Line 413: Please replace “is” to “was”. 

Line 495 – 497: Three things are listed here. What do you mean by "Rabies events"? Are these potential rabies cases or are they vaccination clinics? Please clarify. 

Line 506 and 507: This is a section heading, whilst line 507 is a sub-heading – the sizing of this is the same and therefore makes it challenging to understand. Please ensure that this is clear for the final editorial process to distinguish the two clearly.

Line 543: “The Strategy teams partly followed CBC-MDV manual…” Please include “the” or make “Manual” plural. 

Line 609: “and increase community support and contributions” Please include the word “can” or a similar grammatical revision of the sentence. 

Line 619: It is mentioned that “information recording” was carried out with high-fidelity, yet in lines 500-502, it is highlighted that reporting load was too high and not done according to protocol. I assume that “information recording” relates to the “reporting on dogs vaccinated” – if this is the case, then please specify. 

Line 673 – 674: “account for 3 out of the 7 missed rounds by all strategies and recording a lower coverage at month 11.” - The sentence is a little ambiguous as it relates to the comparison with other strategies but can also be interpreted as a lower coverage when compared with month 1 of strategy III (as the authors assumedly intended). If a comparison of the vaccination coverages in month 11 is done, Strategy III is only lower than Strategy II. It is higher than Strategy I and the Pulse strategy. 

Line 711: Please correct “Delivery” to “Deliver”. 

Line 723: The word “dose” has no context. Are the authors referring to the delivery of more vaccine doses? Please consider and clarify. 

Supplementary material:

Table S2, sub-heading “Providing continuous access to dog vaccination”, first row: The total listed here is 36. Please correct. 

Table S2, sub-heading “Delivery of free dog vaccination clinics using suitable approaches”, first row: Spelling mistake “Start”. 

Table S3: Please clarify as to what “A-day” is?

Table S3: 9 wards are mentioned here of the 12 wards from each district (i.e., 36 wards across 3 districts?). I assume the other 3 wards for this particular district were the Pulse campaign? When was that campaign undertaken and how? Where are the data for the other 2 Districts? (refer to lines 164 - 167 in the main text and the earlier comment).

Reviewer #2: Specific comments (with line numbers):

33. Suggest changing to “can eliminate rabies in dogs,”…

40. Define UK MRC

48-49. Should this be in the methods rather than the results?

69-85. The abstract discusses the 40% coverage level, but the author summary discusses the 70% coverage level. This confused this reviewer initially.

165. A map or visual showing the structure of the wards, villages, and sub-villages could be useful if this would not breach ethics permits. It would be useful for readers not familiar with the region to understand the issues faced by the RC and OHC to travel between locations and gain support in geographically and potentially culturally separated groups.

180. Table 1 overlaps with Figure 1, and this reviewer did not find Figure 1 particularly useful, despite generally having a preference for figures over tables. This reviewer suggests to consider if Figure 1 is needed.

207. Table 2. Remove comma from “24, fortnightly meetings”

317. Figure 2 also has significant overlap with a table, and there was no legend with the figure.

321. Can Table 1 (supplementary file) be renamed to Table S1?

345. The need to longer lead times should be a discussion point (maybe it already is?).

359-364. This seems like it could be a major problem in repeated studies and in larger studies. Will working more directly with the mabalozis really solve the problem, or will some of them think that their social status entitles them to compensation without putting in an equivalent effort to the OHCs in this study?

412-414. How did the cracked pots affect vaccine effectiveness? Showing the temperature data from the pots would be helpful for understanding if this low-tech approach is feasible for large-scale vaccination campaigns.

442-444. This seems strange than none of the OHCs gave out their mobile phone number and a large oversight not to have it included with information packets and advertising. A simple business card for the OHCs could have also partially solved this problem.

442-480. The issue with aggressive dogs appears to be a major issue. It would be good to add a discussion section as to why oral rabies vaccines were not used, and/or whether they could be more effective in this situation.

581-584 (Table 6). Table 6 is very challenging to interpret and has several open boxes. Please consider a complete redesign of this table to more effective convey the information. A take home message from Table 6 seems to be that people did not follow the instructions. A major question left at the end of this study is why should we expect a difference in subsequent trials? Several useful suggestions are offered, such as incorporating the vaccination dates into the local calendars, which could then lead to the vaccination dates becoming standardised over time. However, this information gets swamped in this complex table.

The comment that the OHCs “thought did more work than agreed upon” is concerning. Future studies will need to be very explicit in expected quantitative outputs to reduce the risk of alienating key partners and local vaccination champions.

589. At what level is the “did not include direct involvement of community members” comment aimed? Does this refer to a person in each sub-village or more generally not working close enough with the community or mabolozis.

616-617. Are these references for dog vaccine studies?

638. Are RCs unpopular or not respected in the community due to collecting levies at auctions? I can’t think of many situations where people are happy to pay taxes/levies, so using an RC associated with levies might not be a good strategy.

646-667. This is a very long paragraph that could be trimmed substantially.

689-694. Very long second sentence. Split to simplify.

708-724. This was a fantastic study with a great level of detail included in the manuscript, but was challenging to read at times. This conclusion needs to be much simpler so that anyone that reads part or all of the document retains and simple take away story.

**Summary and General Comments**

Reviewer #1: The manuscript describes the implementation of a community-based continuous mass dog vaccination (CBC-MDV) methodology in select rural districts in Tanzania as a means to evaluate their efficacy towards achieving a sustained vaccination coverage of 40% throughout the course of a year – a suitable coverage to control and eventually eliminate rabies in the target areas. This approach is in contrast to the typical “annual mass dog vaccination” (referred to as the Pulse approach) that is typically followed by national governments and stakeholders as a means to achieve rabies control. The manuscript clearly details the various aspects towards the implementation of this approach as well as the limitations, and adjustments, that were required to ensure effective implementation. By following a detailed and comprehensive process and by carefully planning the study, the impacts of the various CBC-MDV strategies were evaluated and lessons learned for the subsequent proposed scale-up of the method to other areas in the country.

I would like to commend the authors on a comprehensive study, especially considering the scale and logistical requirements to undertake such a study. 

Overall, the manuscript is clear and well-written, albeit a little lengthy. However, this is understandable considering the need to convey the various key aspects of the data and outcomes, and lessons learned. While the study itself was well implemented, there are some aspects of the manuscript that require some thought. 

My main comment was whether the manuscript was aimed at simply determining the most effective CBC-MDV approach, or whether it was aimed as a comparison with the “Pulse” vaccination approach. Throughout the manuscript, the primary focus is on the three CBC-MDV strategies and determining which of those is most effective. Yet, there are some references to the pulse vaccination approach. It is not clear as to whether the manuscript simply aims to highlight (through the detailed methodology) the best CBC-MDV approach and demonstrate that this approach is indeed effective to control rabies, or whether it aims to highlight that the CBC-MDV approach is more effective than the pulse vaccination. If it is the latter, then I believe that there need to be more comparisons between the pulse and CBC-MDV approaches (e.g., in advertising intensity, effort spent on the campaign, at what level and the methodology for the pulse campaign, etc.), including more detail about how the pulse was implemented and what the results were. If it is the former, then I would suggest removing reference to the pulse campaigns, as this simply confuses the aim of the manuscript. Some of the general comments below also refer to this question. 

General Comments:

1) It would be helpful to have a better understanding of the scale of the various administrative levels. For instance, how big is a ward (on average), and then within that, how big is a village, sub-village. What is the distance between sub-villages? Later in the results for instance, the authors mention that the OHC did not work with the Mabalozi who are responsible for 10 households, and rather worked with sub-village leaders. Approximately how many households are in a sub-village – would this be two mabalozi, or would it be 20? This could perhaps have an influence as to why the sub-village leaders were rather approached (in addition to the reasons provided). This would also help to clarify the challenges and fatigue faced by RCs and OHCs when working at the sub-village level, as the distance between sub-villages for example, have not been shared with the reader. 

2) How many animals were vaccinated in each sub-village/village/ward using the different methods. While vaccination coverage is a good indicator in terms of controlling the disease, it is not suitably descriptive when comparing the efficacy of different methods, as the dog population in one village/ward may be significantly greater than in another, thus skewing the interpretation of the results.

3) One of the main limitations listed for a “Pulse” vaccination campaign is the logistical costs and need for vehicles and fuel to deliver the annual vaccination. However, the CBC-MDV method relied on Ward-level livestock field officers (and in theory the district vet) who would need to travel to each village/sub-village to undertake the vaccinations. How was this done (what mode of transportation was used; a reference to walking and use of a motorbike was mentioned when describing the advertising campaign) and what were the costs associated with this compared to the pulse vaccination? I would assume that while these would typically be shorter trips, they would need to be undertaken more frequently, especially if on-demand vaccination is offered. While I understand that this was a compromise from the initial concept, the fact that law only permits vaccination by recognized staff is a common challenge in most rabies-endemic countries. This limitation is emphasized in lines 337 – 338 citing the lack of supervision from District veterinary officers as well as in lines 397-398 citing the long distances required to travel between villages for the advertising campaign. 

4) Another question relating to the limitations of a pulse campaign versus CBC-MDV was the timing of the campaign – yet the CBC-MDV strategies were equally affected in terms of funerals, other commitments, poor weather, etc. If a Pulse campaign is well planned, these limitations can be easily avoided. 

5) Line 310 – 313 mentions one ward from each district undertaking each of the three CBC-MDV strategies, while one ward from each district undertook the pulse campaign. However, in line 164 – 167, the authors mention that the pilot phase included 12 wards from each of the 3 districts (i.e., 36 wards in total). Please can you clarify the reasons as to why these figures differ. If the study was undertaken in 3 representative wards (one from each district) for each strategy, then this needs to be clarified in the methodology. 

6) Line 401: For the advertising campaigns, 56% of OHC’s varied the advertising time to either morning or afternoon to attempt to reach more people. Was it the case that more people were reached at these times, or was this simply due to the inconvenience/other factors relating to deviating from the protocol? i.e., In those villages where the announcements were done at the "varied" time, were vaccination turnout rates lower?

7) Table 5 shows the vaccination coverage for the pulse versus the 3 CBC-MDV strategies. While the pulse campaign achieved far lower coverage, was this as a result of poor planning and implementation rather than due to a reduced efficiency? The drop in vaccination coverage from the pulse campaign was only approximately 3% over the course of the year (assumedly due to births/new dogs). If the initial pulse campaign had reached the 70% (or more) coverage, then the overall vaccination coverage would not have decreased significantly, and the disease could be easily controlled.

Reviewer #2: This manuscript describes an innovative process of a community-based continuous mass dog vaccination (CBC-MDV) process. The key advance of the manuscript was integrating the research project and rabies control measures into the community. The use of qualitative thematic analysis along with more traditional quantitative metrics such as vaccination coverage provides a depth of analysis not often seen in vaccine studies, particularly for non-human studies. The project encountered many challenges, and the process needs further optimisation before large scale implementation. However, this study represents a leap forward in the way free-roaming animals can be vaccinated. I found myself making many notes about how this could be applied to my own research project, which is a good sign for potential impact of the process described here. I strongly recommend this article for publication pending thorough revision. 

A major comment is that the readability of the tables and text needs to be improved. Several parts of the paper appear to be redundant, such as stating very similar information in the methods and results. The formatting of several tables makes it challenging to cross-reference information in separate columns; please do not let Tables break across pages, as this made Tables 2-3 difficult to interpret. Additionally, the Supplementary File contains three tables, but they are not titled as supplementary tables (Table S1, Table S2, Table S3), which leads to confusion about the tables in the main manuscript. The figures at the end of the main manuscript are in the order 1, 3, 4, 5, 2. Figure 2 has no legend and the legends for the other figure are inadequate. A general comment throughout is that sentences should be shortened, and paragraphs should have at least three sentences. There are many non-standard acronyms in this manuscript; a short section at the beginning of the manuscript listing all the acronyms would be useful.

This strategy could be useful for many non-human diseases. It would be good to discuss how this approach could be used for non-rabies pathogens, and even for oral rabies vaccination campaigns for wildlife (fox, raccoon).

PLOS authors have the option to publish the peer review history of their article (what does this mean?). If published, this will include your full peer review and any attached files.

Reviewer #1: Yes: Terence Scott

Reviewer #2: No

Figure Files:

Data Requirements:

Reproducibility:

References

---

## [Editor Report · Decision Letter 1]

22 Aug 2022

Dear Mr. Duamor,

We are pleased to inform you that your manuscript 'Development, feasibility and potential effectiveness of community-based continuous mass dog vaccination delivery strategies: lessons for optimization and replication' has been provisionally accepted for publication in PLOS Neglected Tropical Diseases.

Best regards,

David Joseph Diemert, M.D.

Academic Editor

Gregory Moseley

Section Editor

---

## [Editor Report · Acceptance letter]

31 Aug 2022

Dear Mr. Duamor,

We are delighted to inform you that your manuscript, "Development, feasibility and potential effectiveness of community-based continuous mass dog vaccination delivery strategies: lessons for optimization and replication," has been formally accepted for publication in PLOS Neglected Tropical Diseases.

Best regards,

Shaden Kamhawi

co-Editor-in-Chief

Paul Brindley

co-Editor-in-Chief
